# Effects of Chitosan Oligosaccharide on Plasma and Hepatic Lipid Metabolism and Liver Histomorphology in Normal Sprague-Dawley Rats

**DOI:** 10.3390/md18080408

**Published:** 2020-08-02

**Authors:** Shing-Hwa Liu, Rui-Yi Chen, Meng-Tsan Chiang

**Affiliations:** 1Graduate Institute of Toxicology, College of Medicine, National Taiwan University, Taipei 10051, Taiwan; shinghwaliu@ntu.edu.tw; 2Department of Medical Research, China Medical University Hospital, China Medical University, Taichung 40402, Taiwan; 3Department of Pediatrics, College of Medicine, National Taiwan University Hospital, Taipei 10051, Taiwan; 4Department of Food Science, National Taiwan Ocean University, Keelung 20224, Taiwan; v33322733@gmail.com

**Keywords:** chitosan oligosaccharide, normal rats, lipid metabolism, liver histomorphology

## Abstract

Chitosan oligosaccharide is known to ameliorate hypercholesterolemia and diabetes. However, some studies found that chitosan oligosaccharide might induce mild to moderate hepatic damage in high-fat (HF) diet-induced obese rats or diabetic rats. Chitosan oligosaccharide can be as a dietary supplement, functional food, or drug. Its possible toxic effects to normal subjects need to be clarified. This study is designed to investigate the effects of chitosan oligosaccharide on plasma and hepatic lipid metabolism and liver histomorphology in normal Sprague-Dawley rats. Diets supplemented with 5% chitosan oligosaccharide have been found to induce liver damage in HF diet-fed rats. We therefore selected 5% chitosan oligosaccharide as an experimental object. Rats were divided into: a normal control diet group and a normal control diet +5% chitosan oligosaccharide group. The experimental period was 12 weeks. The results showed that supplementation of 5% chitosan oligosaccharide did not significantly change the body weight, food intake, liver/adipose tissue weights, plasma lipids, hepatic lipids, plasma levels of AST, ALT, and TNF-α/IL-6, hepatic lipid peroxidation and anti-oxidative enzyme activities, fecal lipids, and liver histomorphology in normal rats. These findings suggest that supplementation of 5% chitosan oligosaccharide for 12 weeks may not induce lipid metabolism disorder and liver toxicity in normal rats.

## 1. Introduction

Chitosan oligosaccharide, which is a product that is mainly prepared from chitin or chitosan by acid hydrolysis or enzyme hydrolysis, is characterized by low molecular weight (MW), high degree of deacetylation (DD), low degree of polymerization (DP), less viscous, and complete water solubility [1]. Its DD is >90%, DP is <20, and average MW is <3900 Da [1,2]. However, Muanprasat and Chatsudthipong (2017) have defined that chitosan oligosaccharide has a DP < 50–55 and an average MW < 10,000 Da [3]. It has good water solubility and biocompatibility, and is easily absorbed in the human body. Chitosan oligosaccharide can be absorbed through intestinal epithelial cells, and an increase in its absorption rate is accompanied by a decrease in the average molecular weight [1,3]. Many studies have indicated that chitosan oligosaccharide possesses many pharmacological functions, such as anti-inflammation [4], anti-cancer [5], anti-diabetes [6], anti-bacteria [7], and anti-oxidation [8]. Chitosan oligosaccharide can also regulate lipid metabolism, improving obesity and dyslipidemia [9,10,11]. Chitosan oligosaccharide has been shown to inhibit adipocyte differentiation and reduce lipid accumulation [12], improve obesity-related glucolipid metabolism disorder in mouse liver [13], and attenuate nonalcoholic fatty liver in mice [14].

Naveed et al. (2019) have recently mentioned that the application of chitosan derivatives and chitosan oligosaccharide as a dietary blend is still a controversial issue by regulatory authorities because of lacking extensive safety data [1]. They further indicated that the impurities or degradation products from chitosan oligosaccharide and its derivatives may be the risk factors [1]. Some previous studies have shown that chitosan oligosaccharide with various average MW and DD causes varying degrees of adverse effects in vitro and in vivo [15,16,17,18]. Teodoro et al. (2016) have found that treatment with 0.5% chitosan oligosaccharide in drinking water for six weeks significantly reduces the body weight in normal control Wistar rats, but increases the levels of AST, total bilirubin, and direct bilirubin in Goto-Kakizaki diabetic rats (a non-obese type 2 diabetic animal model), which might indicate mild hepatic injury [15]. They suggested that chitosan oligosaccharides may have potential for toxicological side effects in a diabetic condition. Moreover, Chiu et al. (2019) have recently shown that diets supplemented with 5% chitosan oligosaccharide in high-fat diet-induced obese rats for eight weeks significantly and markedly increased plasma levels of AST, ALT, and tumor necrosis factor-alpha (TNF-α), suggesting that supplementation of 5% chitosan oligosaccharide may induce liver injury in an obese condition [10]. The effects of 5% chitosan oligosaccharide on rats with a normal condition for the longer period of supplementation are still unknown. Its possible toxic effects for long-term administration to normal subjects need to be clarified. Therefore, this study is designed to investigate the long-term effects of 5% chitosan oligosaccharide in diets on plasma and hepatic lipid metabolism and liver histomorphology in normal Sprague-Dawley rats.

## 2. Results and Discussion

### 2.1. Effects of Chitosan Oligosaccharide on Body and Tissue Weights and Food Intake in Normal Rats

Male Sprague-Dawley rats fed the normal control diet (NC group) or normal control diet +5% chitosan oligosaccharide (CC group) for 12 weeks were used in the experiments. As shown in Figure 1 and Table 1, there were no statistically significant changes in body weight and body weight gain between the NC group and CC group. Supplementation of chitosan oligosaccharide could also not change the food intake and food efficiency in normal rats compared to the NC group (Table 1). Moreover, there were no statistically significant differences in the liver and adipose tissue weights between the NC group and CC group (Table 2).

Normal Wistar rats treated with 0.5% chitosan oligosaccharide in drinking water for six weeks have been found to have significantly reduced body weight, but not liver weight [15]. Zhang et al. (2019) recently reported that normal Sprague-Dawley rats intragastrically treated with 50 mg/kg chitosan oligosaccharide for two weeks did not change body weight and heart weight [19]. Kim et al. (2009) have also shown no changes in body weight in normal Sprague-Dawley rats treated with chitosan oligosaccharide at a dosage of 500 mg/kg by oral gavage for eight weeks [20]. The 5% chitosan oligosaccharide used in the present study is approximately equal to 5000 mg/kg (rat with 300 g of body weight and 30 g of daily food intake). Our results are consistent with the findings of Kim et al. (2009) and Zhang et al. (2019), although there are differences in chemical characterization, dosage, administration route, and experimental period between these two studies.

### 2.2. Effects of Chitosan Oligosaccharide on the Levels of Plasma, Hepatic, Adipose, and Fecal Lipids in Normal Rats

As shown in Table 3, for the levels of plasma glucose, total cholesterol (TC), and triglyceride (TG), there were no statistically significant changes between the CC and NC groups. Similarly, the TC and TG levels in the livers were also not changed in normal rats fed diets with 5% chitosan oligosaccharide compared to the NC group (Table 3). Moreover, the levels of TG in the adipose tissues were also not changed in the CC group compared to the NC group (Figure 2). On the other hand, we examined the changes in fecal lipid metabolism. As shown in Table 4, there were no differences in the fecal weights and the levels of fecal TC and TG between the NC group and CC group.

Teodoro et al. (2016) found that normal Wistar rats treated with 0.5% chitosan oligosaccharide in drinking water for six weeks did not change the levels of plasma glucose and TG, but significantly reduced plasma TC levels [15]. Zhang et al. (2019) recently showed that normal Sprague-Dawley rats intragastrically treated with 50 mg/kg chitosan oligosaccharide for two weeks did not affect the levels of serum TC and HDL-C [19]. Kim et al. (2009) also reported that normal Sprague-Dawley rats treated with chitosan oligosaccharide (500 mg/kg) by oral gavage for eight weeks did not change the levels of plasma glucose and TG, but slightly elevated plasma TG [20]. These previous studies and our study suggest that administration of chitosan oligosaccharide to normal rats may not be interfering with plasma lipid metabolism, although there are differences in chemical characterization, dosage, administration route, and experimental period among these studies. We further demonstrated that long-term administration of 5% chitosan oligosaccharide to normal rats did not affect hepatic lipid metabolism.

### 2.3. Effects of Chitosan Oligosaccharide on the Biochemical Parameters, Histology, and Liver Lipid Peroxidation and Anti-Oxidative Enzyme Activities in Normal Rats

We next investigated the long-term effects of 5% chitosan oligosaccharide on liver function markers, liver histomorphology, and liver lipid peroxidation in normal rats. As shown in Figure 3, normal rats fed diets supplemented with 5% chitosan oligosaccharide for 12 weeks did not change the AST and ALT activities, TNF-α level, and IL-6 level in plasma compared to the NC group. There were no statistically significant differences in the TBARS level and SOD and GPx activities between the NC group and CC group (Table 5). Moreover, the results of histological analysis also showed that there were no obvious differences in liver histomorphology between the NC group and CC group (Figure 4).

Kim et al. (2001) have found that there are no observed toxic effects in rats treated with chitosan oligosaccharide (average MW < 1000 Da; 500–2000 mg/kg) for four weeks [21]. Teodoro et al. (2016) have indicated that treatment with 0.5% chitosan oligosaccharide (average MW 5000 Da; DD > 90%) in drinking water for six weeks significantly increases the AST activity in the plasma of diabetic rats, but it does not change the AST and ALT activities in the plasma of normal control rats [15]. Zhang et al. (2019) have also found that rats intragastrically treated with 50 mg/kg chitosan oligosaccharide (average MW 5000 Da; DD > 90%) for two weeks were significantly protected against doxorubicin-increased lipid peroxidation and -decreased anti-oxidative enzyme activities in the heart, but it does not change the lipid peroxidation and anti-oxidative enzyme activities in the hearts of normal control rats [19]. The 5% chitosan oligosaccharide used in the present study is approximately equal to 5000 mg/kg. These previous studies and our study suggest that the plasma AST and ALT activities, hepatic lipid peroxidation, and anti-oxidative enzyme activities are not changed by administration of chitosan oligosaccharide to normal rats, although there are differences in dosage, administration route, and experimental period among these studies. We further demonstrated that long-term administration of 5% chitosan oligosaccharide to normal rats could also not affect the inflammatory markers TNF-α and IL-6 levels in the plasma and histological morphology in the liver.

## 3. Materials and Methods

### 3.1. Chemicals

Chitosan oligosaccharide was purchased from Koyo Chemical Co. Ltd. (Tokyo, Japan). The average MW and degree of deacetylation (DD) of chitosan oligosaccharides were about 719 Dalton and 100%, respectively. 1,1,3,3,-tetraethoxypropane was obtained from Sigma-Aldrich (St. Louis, MO, USA). The TC and TG enzymatic assay kits were obtained from Audit Diagnostics (Cork, Ireland). The AST and ALT enzymatic assay kits were provided by Randox Laboratories (Antrim, UK). The reagents for hematoxylin and eosin stain were purchased from Leica Biosystems (Richmond, IL, USA). The glucose enzymatic kit was purchased from Randox Laboratories. The rat TNF-α and IL-6 enzyme immunometric assay kits were provided by R&D systems, Inc. (Minneapolis, MN, USA). The SOD assay kit was purchased from Cayman Chemical (Ann Arbor, MI, USA). The GPx assay kit was obtained from Cayman Chemical.

### 3.2. Animals

The animal experiments were approved by the Animal House Management Committee of the National Taiwan Ocean University. The managements for animal experiments were in accordance with the guidelines for the care and use of laboratory animals [22]. Six-week-old male Sprague-Dawley (SD) rats were obtained from BioLASCO (Taipei, Taiwan). Rats were housed in cages maintained at 23 ± 1 °C and 40–60% relative humidity with a 12 h light/12 h dark cycle. Rats were given the standard laboratory diets (5001 rodent diet, LabDiet, St. Louis, MO, USA) and deionized water ad libitum. After a 1-week acclimation period, rats were randomly divided into two groups: normal control diet group and normal control diet +5% chitosan oligosaccharide group. The diet composition is shown in Table 6. The experiments were conducted for a total of 12 weeks. Food intake was measured every 3 days, and body weight was measured once a week. Feces were collected on the last three days of week 12, and then feces were dried and weighed.

### 3.3. Sampling Blood and Tissue

The samples of blood, liver, and perirenal and para-epididymal adipose tissues were collected after euthanization of rats under anesthesia at the end of the experiment. Plasma was prepared by centrifugation at 1750× *g* for 20 min (4 °C). These samples were instantly frozen and stored at −80 °C until further analysis.

### 3.4. Measurements of Blood Glucose, Plasma Lipids, Lipoproteins, Activities of Aspartate Aminotransferase (AST) and Alanine Aminotransferase (ALT), Liver and Fecal Lipids, Plasma TNF-α and IL-6, Liver Superoxide Dismutase (SOD) Activity, Liver Glutathione Peroxidase (GPx) Activity, and Liver Lipid Peroxide (Thiobarbituric Acid Reactive Substances, TBARS) Content

The blood glucose was determined by a glucose enzymatic kit (Randox Laboratories). The absorbance at 500 nm was detected by a spectrophotometer (UV/VIS-7800, JASCO International, Tokyo, Japan).

The enzymatic assay kits for TC and TG (Audit Diagnostics) were used to detect the plasma TC and TG levels. The absorbance at 500 nm was determined by a spectrophotometer (UV/VIS-7800, JASCO International). The enzymatic assay kits for AST and ALT (Randox) were used to analyze the activities of AST and ALT. The absorbance at 340 nm was determined by a spectrophotometer (UV/VIS-7800, JASCO International).

Both liver and fecal lipids were extracted according to the method of Folch et al. [23]. The levels of TG and TC in both livers and feces were determined as previously described by Carlson and Goldfarb [24].

The plasma TNF-α and IL-6 were determined by the Rat TNF-α and IL-6 enzyme immunometric assay kit (R&D systems). The absorbance at 450 nm was measured by a VersaMax microplate (Molecular Device, San Jose, CA, USA).

The liver SOD activity was determined by a SOD assay kit (Cayman Chemical). The absorbance at 440 nm was measured by a VersaMax microplate (Molecular Device).

The liver GPx activity was determined by a glutathione peroxidase assay kit (Cayman Chemical). The absorbance at 340 nm was measured by a VersaMax microplate (Molecular Device).

The principle of the method of TBARS detection was to use thiobarbituric acid (TBA) to react with the lipid peroxide product (malondialdehyde, MDA) in the liver to produce color, thereby determining the lipid peroxide content in the liver. The 1,1,3,3,-tetraethoxypropane (Sigma-Aldrich) was the standard and physiological saline was the blank group. The absorbances at 520 and 535 nm were determined by a Hitachi U2800A spectrophotometer.

### 3.5. Examination of Liver Histomorphology

The analysis of liver histomorphology was determined as previously described [25]. The hematoxylin and eosin (H&E) was used to stain liver tissue sections, which were 5-μm thick paraffin sections. The stained tissue sections were then observed and imaged by a photomicroscope (Nikon Eclipse TS100, Nikon Instruments, Melville, NY, USA) equipped with a digital camera (Nikon D5100, Nikon Instruments).

### 3.6. Analysis of Statistics

All experimental data were presented as mean ± standard deviation (S.D.) and were analyzed with Student’s *t* test using the GraphPad Prism v6.0 software (GraphPad Software, San Diego, CA, USA). There was a significant difference between the two groups when *p* was less than 0.05.

## 4. Conclusions

In the present study, the results indicate that supplementation of 5% chitosan oligosaccharide for 12 weeks does not interfere with plasma and hepatic lipid metabolism in normal rats. The lipid peroxidation, anti-oxidative enzyme activities, and histomorphology in the livers are also not changed by 5% chitosan oligosaccharide supplementation in normal rats (the findings were summarized in Figure 4B). These findings suggest that diets supplemented with 5% chitosan oligosaccharide for 12 weeks may not induce lipid metabolism disorder and liver toxicity in normal rats.

## Figures and Tables

**Figure 1 marinedrugs-18-00408-f001:**
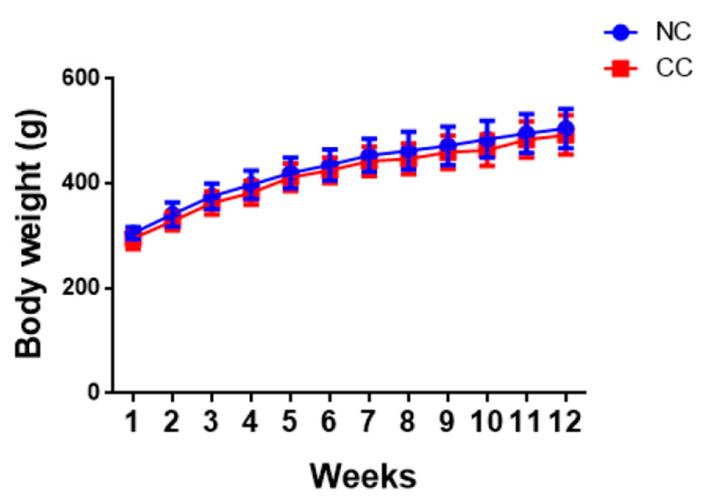
Effects of chitosan oligosaccharide on body weight in normal SD rats for 12 weeks. Results are presented as the mean ± SD for each group (n = 10). NC: normal control diet (chow diet). CC: normal control diet +5% chitosan oligosaccharides.

**Figure 2 marinedrugs-18-00408-f002:**
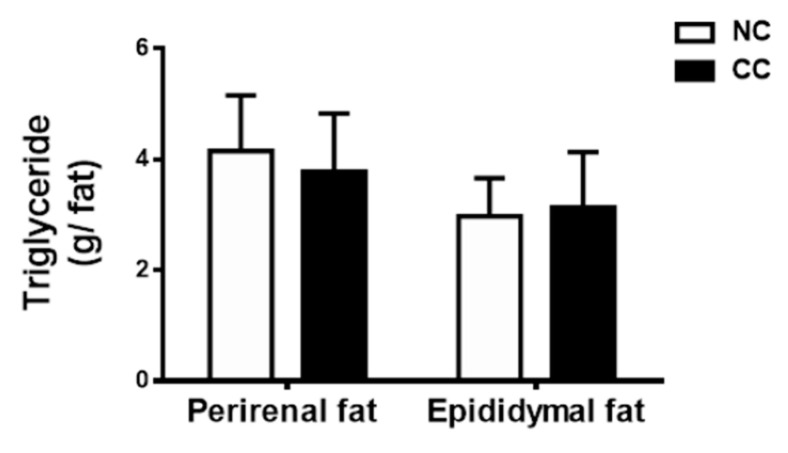
Effects of chitosan oligosaccharide on the levels of adipose tissue triglyceride in normal SD rats for 12 weeks. Results are expressed as the mean ± SD for each group (n = 10). NC: normal control diet (chow diet). CC: normal control diet +5% chitosan oligosaccharides.

**Figure 3 marinedrugs-18-00408-f003:**
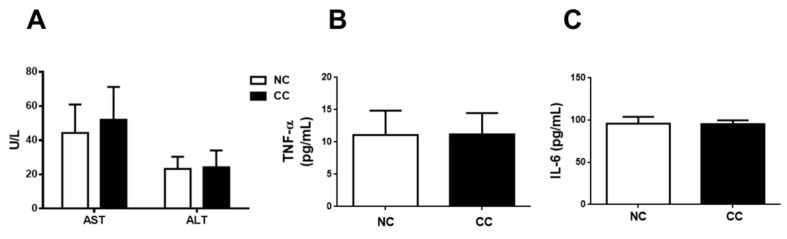
Effects of chitosan oligosaccharide on plasma AST, ALT, and TNF-α in normal SD rats for 12 weeks. The plasma AST and ALT activities (**A**) and TNF-α level (**B**) and IL-6 level (**C**) are shown. Results are expressed as the mean ± SD for each group (n = 10). NC: normal control diet (chow diet). CC: normal control diet +5% chitosan oligosaccharides.

**Figure 4 marinedrugs-18-00408-f004:**
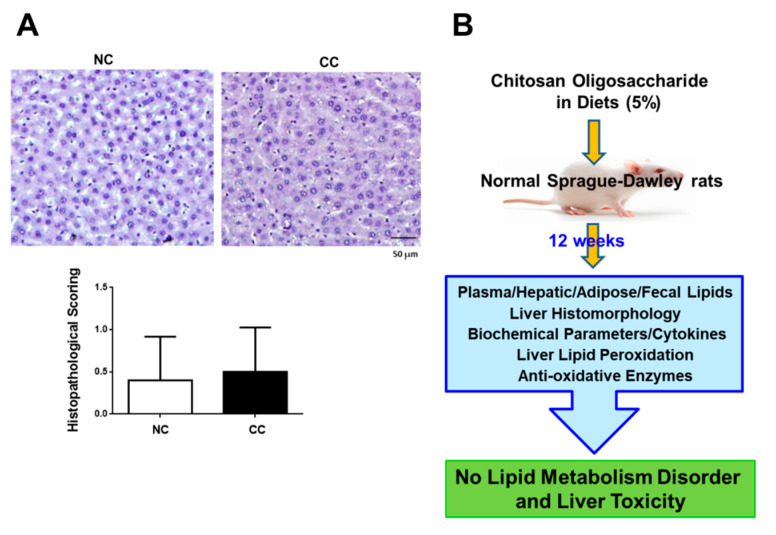
(**A**) The liver histomorphology in normal SD rats fed diets supplemented with chitosan oligosaccharide for 12 weeks. Histological analysis of livers is shown. Tissue sections were stained with hematoxylin and eosin stain. Scale bar = 50 μm. NC: normal control diet (chow diet). CC: normal control diet +5% chitosan oligosaccharides. (**B**) Scheme representing the effects of chitosan oligosaccharide on plasma and hepatic lipid metabolism and liver histomorphology in normal Sprague-Dawley rats.

**Table 1 marinedrugs-18-00408-t001:** Effects of chitosan oligosaccharide on body weight, body weight gain, and food intake in the Sprague-Dawley (SD) rats for 12 weeks.

Parameters	NC	CC
Initial body weight (g)	224.0 ± 5.3	224.1 ± 5.2
Final body weight (g)	489.8 ± 40.1	480.2 ± 36.4
Body weight gain (g)	265.7 ± 37.9	256.1 ± 38.6
Food intake (g/day)	30.8 ± 1.6	30.2 ± 2.1
Food efficiency (%)	8.6 ± 0.9	8.5 ± 1.4

Results are expressed as the mean ± SD for each group (n = 10). NC: normal control diet (chow diet). CC: normal control diet +5% chitosan oligosaccharide.

**Table 2 marinedrugs-18-00408-t002:** Effects of chitosan oligosaccharide on organ/tissue weights in SD rats for 12 weeks.

Parameters	NC	CC
Liver weight (g)	14.22 ± 1.32	13.85 ± 1.63
Relative liver weight (g/100 g BW)	2.90 ± 0.18	2.89 ± 0.36
Perirenal fat (g)	8.37 ± 2.65	7.55 ± 2.47
Relative Perirenal fat weight (g/100 g BW)	1.68 ± 0.44	1.58 ± 0.54
Epididymal fat (g)	6.19 ± 1.26	6.02 ± 1.74
Relative Epididymal fat weight (g/100 g BW)	1.25± 0.21	1.29 ± 0.43
Total adipose tissue weight (g)	14.56 ± 3.82	13.94 ± 4.38
Relative adipose tissue weight (g/100 g BW)	2.94 ± 0.62	2.92 ± 0.97

Result are expressed as the mean ± SD for each group (n = 10). NC: normal control diet (chow diet). CC: normal control diet +5% chitosan oligosaccharides.

**Table 3 marinedrugs-18-00408-t003:** Effects of chitosan oligosaccharide on the levels of plasma glucose and lipids and liver lipids in SD rats for 12 weeks.

Parameters		NC	CC
Plasma	Glucose (mg/dL)	205.16 ± 19.06	201.08 ± 31.23
	Total cholesterol (mg/dL)	58.95 ± 7.15	53.96 ± 4.02
	Triglyceride (mg/dL)	55.83 ± 6.80	52.03 ± 3.19
Liver	Total cholesterol		
	(mg/g liver)	3.76 ± 0.45	3.60 ±0.50
	(g/liver)	0.053 ± 0.002	0.049 ± 0.002
	Triglyceride		
	(mg/g liver)	19.15 ± 1.07	18.53 ± 3.00
	(g/liver)	0.27 ± 0.03	0.26 ± 0.05

Results are expressed as the mean ± SD for each group (n = 10). NC: normal control diet (chow diet). CC: normal control diet +5% chitosan oligosaccharides.

**Table 4 marinedrugs-18-00408-t004:** Effects of chitosan oligosaccharide on fecal weight, total cholesterol, and triglyceride in normal SD rats for 12 weeks.

Parameters	NC	CC
Feces wet weight (g/day)	8.67 ± 1.04	9.11 ± 1.39
Feces dry weight (g/day)	5.97 ± 0.40	5.87 ± 0.39
Total cholesterol		
(mg/g feces)	5.07 ± 0.77	5.11 ± 0.57
(mg/day)	30.26 ± 4.69	30.14 ± 4.76
Triglyceride		
(mg/g feces)	5.63 ± 1.60	5.96 ± 1.85
(mg/day)	33.69 ± 10.11	35.19 ± 12.09

Results are expressed as the mean ± SD for each group (n = 10). NC: normal control diet (chow diet). CC: normal control diet +5% chitosan oligosaccharides.

**Table 5 marinedrugs-18-00408-t005:** Effects of chitosan oligosaccharide on the liver TBARS level and antioxidative enzyme activities in normal SD rats for 12 weeks.

Diet	NC	CC
TBARS (nmole/mg protein)	0.23 ± 0.06	0.24 ± 0.06
SOD (U/mg protein)	2.34 ± 0.47	2.43 ± 0.37
GPx (U/mg protein)	30.22 ± 6.36	28.99 ± 9.04

Results are expressed as the mean ± SD for each group (n = 10). NC: normal control diet (chow diet). CC: normal control diet +5% chitosan oligosaccharides.

**Table 6 marinedrugs-18-00408-t006:** Composition of experimental diets (%).

Ingredient (%)	NC	CC
Chitosan oligosaccharides ^1^	―	5
Lard	―	1.2
Chow diet	100	93.8
Total calories (kcal/100 g)	336.2	336.1
Carbohydrates (% kcal)	57.9	57.3
Protein (% kcal)	28.7	26.9
Fat (% kcal)	13.4	15.8

NC: control diet; CC: control diet + chitosan oligosaccharides. ^1^ The average MW and DD of chitosan oligosaccharides were about 719 Dalton and 100%, respectively.

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
