# Peer review of "Effects of Chitosan Oligosaccharide on Plasma and Hepatic Lipid Metabolism and Liver Histomorphology in Normal Sprague-Dawley Rats"

_marinedrugs, 2020, doi:10.3390/md18080408_

Round 1
Reviewer 1 Report
The authors have addressed the issues, it is acceptable.
Author Response
Reviewer#1:
The authors have addressed the issues, it is acceptable.
Response: We appreciate the reviewer's positive comment
Reviewer 2 Report
Improvements have been made, but this still remains an essentially negative study which could be markedly condensed to a Short Communication.
Author Response
Improvements have been made, but this still remains an essentially negative study which could be markedly condensed to a Short Communication. Comments and Suggestions for Authors
Response: We appreciate the reviewer's positive comment. We agree with the comment that our findings are essentially negative and agree with presentation as a Communication. We have also revised the manuscript according to the suggestion of reviewer.
This manuscript is a resubmission of an earlier submission. The following is a list of the peer review reports and author responses from that submission.
Round 1
Reviewer 1 Report
Liu et al. described long-term effects of chitosan oligosaccharides on plasma, hepatic lipid, and several biochemical markers for liver toxicity in normal SD rats. They concluded that 5% supplementation of chitosan oligosaccharides does not induce any toxicity in normal SD rats with regard to the above mentioned biochemical makers. This is scientifically sound and all the presented results are consistent with their conclusion. However, this reviewer has some issues as follows;
- Is it still need to clarify the possible toxicity of chitosan oligosaccharides currently used as a functional food and dietary supplement? Several researches in animal and clinical trials have already published that chitosan oligosaccharides are safe. What is difference between this and the published results? Just difference in the period of supplementation?
- Different chitosan oligosaccharides with MW and DD value should be tested to verify the toxicity of chitosan oligosaccharides. Without this information the conclusion is not convincing.
Reviewer 2 Report
Several relevant recent papers have not been cited: PMID: 30463189; PMID: 31744059; PMID: 30708011.
The major finding is essentially negative. It could be presented in a single paragraph with one figure (a modified figure 4) as part of a larger project.
Can 12 weeks in rats be defined as long-term for a toxicological study?
The intervention has not been chemically characterised so comparison with other products even of the same name cannot be made.
Please present dose in units of mg/kg based on food intake as 5% in diet cannot be compared with 500 mg/kg.
Table 1 is too precise – body weights cannot be measured accurately to 10mg.
Figure 4: Needs greater magnification and also quantitative analysis such as NIH Image J program; further staining for lipids (oil red O) and immunohistochemistry for inflammatory cells is essential.